# Outlier Dimensions Encode Task-Specific Knowledge

**William Rudman[1], Catherine Chen[1], Carsten Eickhoff[2]**
Brown University[1]  University of Tübingen[2]
{william_rudman, catherine_s_chen}@brown.edu
carsten.eickhoff@uni-tuebingen.de

## Abstract

Representations from large language models (LLMs) are known to be dominated by a small subset of dimensions with exceedingly high variance. Previous works have argued that although ablating these *outlier dimensions* in LLM representations hurts downstream performance, outlier dimensions are detrimental to the representational quality of embeddings. In this study, we investigate how fine-tuning impacts outlier dimensions and show that 1) outlier dimensions that occur in pre-training persist in fine-tuned models and 2) a single outlier dimension can complete downstream tasks with a minimal error rate. Our results suggest that outlier dimensions can encode crucial task-specific knowledge and that the value of a representation in a single outlier dimension drives downstream model decisions.[1]

## 1 Introduction

Large Language Models (LLMs) are highly over-parameterized with LLM representations utilizing only a small portion of the available embedding space uniformly (Gordon et al., 2020; Prasanna et al., 2020; Rudman et al., 2022). Representations of transformer-based LLMs are dominated by a few *outlier dimensions* whose variance and magnitude are significantly larger than the rest of the model's representations (Timkey and van Schijndel, 2021; Kovaleva et al., 2021). Previous studies devoted to the formation of outlier dimensions in pre-trained LLMs suggest that imbalanced token frequency causes an uneven distribution of variance in model representations (Gao et al., 2019; Puccetti et al., 2022). Although many argue that outlier dimensions "disrupt" model representations, making them less interpretable and hindering model performance, ablating outlier dimensions has been shown to cause downstream performance to decrease dramatically (Kovaleva et al., 2021; Puccetti et al., 2022).

There currently is little understanding of how fine-tuning impacts outlier dimensions and why ablating outlier dimensions is harmful to downstream performance. We address this gap in the literature by investigating 1) how fine-tuning changes the structure of outlier dimensions and 2) testing the hypothesis that outlier dimensions contain task-specific knowledge. This study makes the following novel contributions:

1. We find that outlier dimensions present in pre-training remain outlier dimensions after fine-tuning, regardless of the given downstream task or random seed.
2. We demonstrate that outlier dimensions in ALBERT, GPT-2, Pythia-160M, and Pythia-410M encode task-specific knowledge and show that it is feasible to accomplish downstream tasks by applying a linear threshold to a single outlier dimension with only a marginal performance decline.

## 2 Related Works

Two seminal works discovered the presence of "outlier" (Kovaleva et al., 2021) or "rogue" (Timkey and van Schijndel, 2021) dimensions in pre-trained LLMs. Following Kovaleva et al. (2021) and Puccetti et al. (2022), we define outlier dimensions as dimensions in LLM representations whose variance is at least 5x larger than the average variance in the global vector space. The formation of outlier dimensions is caused by a token imbalance in the pre-training data with more common tokens having much higher norms in the outlier dimensions compared to rare tokens (Gao et al., 2019; Puccetti et al., 2022). Although the community agrees on the origin of outlier dimensions, their impact on the representational quality of pre-trained LLMs has been widely contested.

The concept of isotropy (i.e., the uniformity of variance in a distribution) is closely related to out-

---

[1]Code: *https://github.com/wrudman/outlier_dimensions*

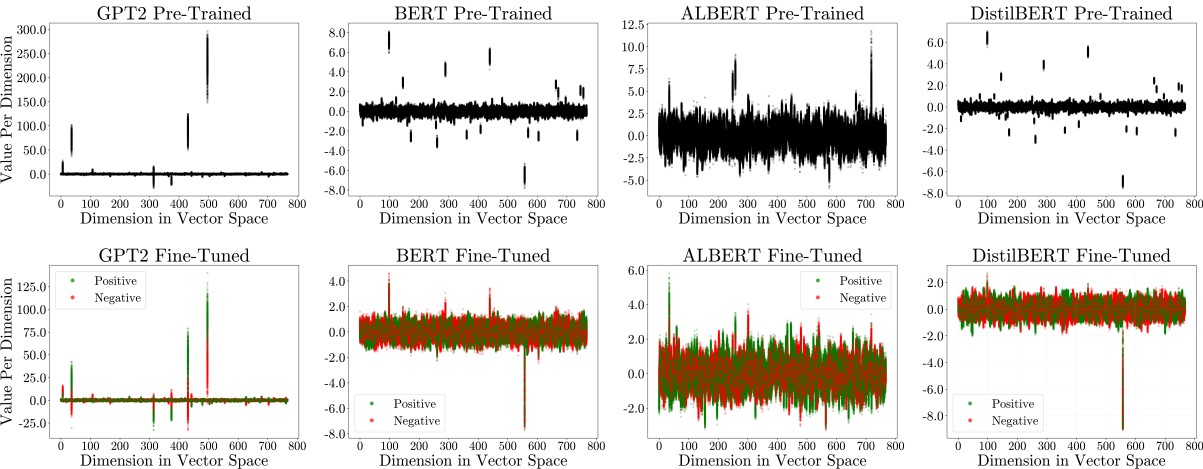

Figure 1: Average activation diagrams of sentence embeddings on the SST-2 validation dataset. The x-axis represents the index of the dimension, and the y-axis is the corresponding magnitude in that given dimension. Top: pre-trained models where no fine-tuning occurs. Bottom: models fine-tuned to complete SST-2.

lier dimensions. Namely, the presence of outlier dimensions causes model representations to be highly *anisotropic* (Rudman and Eickhoff, 2023). Many previous works have argued that mitigating the impact of outlier dimensions by forcing LLM representations to be isotropic improves model interpretability and performance (Rajaee and Pilehvar, 2021; Liang et al., 2021; Mu et al., 2017; Zhou et al., 2020; Gao et al., 2019). Further, Timkey and van Schijndel (2021) claim that outlier dimensions do not meaningfully contribute to the model decision-making process and that removing outlier dimensions aligns LLM embeddings more closely to human similarity judgments. Although the notion that isotropy is beneficial to model representations has been widely adopted in the literature, recent studies have shown that many tools used to measure isotropy and the impact of outlier dimensions in NLP are fundamentally flawed (Rudman et al., 2022).

There is a growing body of literature arguing that anisotropy is a natural consequence of stochastic gradient descent and that compressing representations into low-dimensional manifold correlates with improved downstream performance (Zhu et al., 2018; Ansuini et al., 2019; Recanatesi et al., 2019; Rudman and Eickhoff, 2023). Recent works in NLP suggest that LLMs store linguistic and task-specific information in a low-dimensional subspace (Coenen et al., 2019; Hernandez and Andreas, 2021; Zhang et al., 2023). Further, Rudman and Eickhoff (2023) argue that encouraging the formation of outlier dimensions in LLM representations improves model performance on down-

stream tasks. In this study, we demonstrate that certain LLMs store task-specific knowledge in a *1-dimensional subspace* and provide evidence supporting claims that outlier dimensions are beneficial to model performance.

## 3 Experiments

**Training Details**    We fine-tune 4 transformer encoder LLMs: BERT (Devlin et al., 2018), ALBERT (Lan et al., 2020), DistilBERT (Sanh et al., 2020), RoBERTa (Liu et al., 2019) and 4 transformer decoder LLMs: GPT-2 (Radford et al., 2019), Pythia-70M, Pythia-160M, Pythia-410M (Biderman et al., 2023). We fine-tune our models on 5 binary classification tasks contained in the GLUE benchmark (Wang et al., 2018): SST-2 (Socher et al., 2013), QNLI (Rajpurkar et al., 2016), RTE (Dagan et al., 2005), MRPC (Dolan and Brockett, 2005), QQP. A detailed description of each task is available in Section A of the Appendix. A detailed description of our hyperparameter search, exact hyperparameters, and random seeds is given in Section B. We follow the common practice of reporting results on GLUE tasks using the hidden validation dataset.

### 3.1 Persistence of Outlier Dimensions

**Methods**    After training the model, we calculate the variance of sentence embeddings on the validation data on each task and each random seed and count the number of times a given dimension has a variance 5x larger than the overall average. We visualize outlier dimensions by creating "activation diagrams" where the x-axis is the index of a given dimension, and the y-axis is the magnitude of a

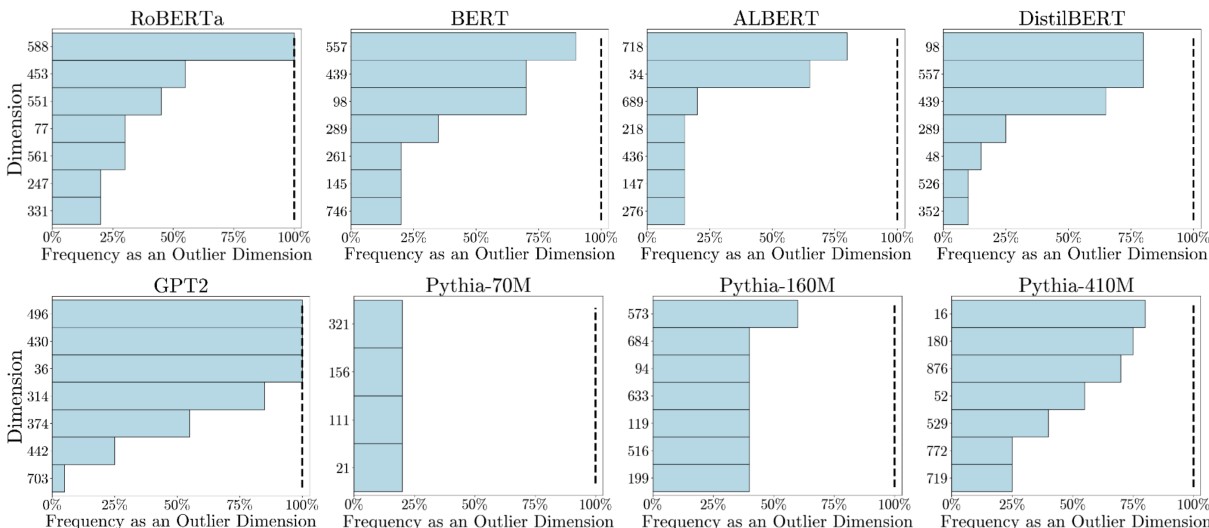

Figure 2: Frequency of the 7 most commonly occurring outlier dimensions across all fine-tuning tasks and all random seeds. The x-axis plots the dimension frequency, and the y-axis plots the dimension index.

sentence embedding in that dimension. We report the average sentence embeddings across 4 random seeds for all activation diagrams.

**Results** Figure 1 demonstrates how fine-tuning impacts model representations of sentence embeddings by plotting activation diagrams from BERT, ALBERT, DistilBERT, and GPT-2 on the SST-2 validation data before (top row) and after (bottom row) fine-tuning. Activation diagrams for the remaining four models are available in Section E in the Appendix. The magnitudes of outlier dimensions in GPT-2 are far larger than any of the models considered in this study. The outlier dimension with the largest variance in GPT-2 has an average variance value of 3511.82 compared to 4.87, 9.30, and 4.68 for BERT, ALBERT, and DistilBERT (see Section C for full results). For GPT-2, fine-tuning exacerbates the variance of existing outlier dimensions but decreases the mean value of outlier dimensions. Notably, in GPT-2, the exact set of top 3 outlier dimensions in pre-training persist when fine-tuning models to complete downstream tasks. Figure 2 demonstrates that a small subset of outlier dimensions emerge for a given model regardless of the downstream classification tasks or the random seed. In particular, in GPT-2 and RoBERTa, there are dimensions that qualify as outlier dimensions for every fine-tuning task and random seed. Outlier dimensions in the Pythia models have a far lower occurrence rate than any of the models in the paper. This finding is especially pronounced in Pythia-70M and Pythia-160M, where no dimensions have

an occurrence rate higher than 70%.

Not only do the outlier dimensions in Pythia models have low occurrence rates, but the Pythia models have far fewer outlier dimensions present in the embedding space compared to BERT, AL-BERT, DistilBERT, and RoBERTa. In general, far more outlier dimensions emerge in the encoder models considered in this study compared to the decoder models. In particular, GPT-2 and Pythia-70M only have 8 and 4 unique outlier dimensions that appear across all fine-tuning tasks and random seeds compared to 62, 60, 24, and 64 for BERT, ALBERT, DistilBERT, and RoBERTa, respectively. Interestingly, Figure 2 shows that the 4 most common outlier dimensions in BERT and DistilBERT are the same, indicating that outlier dimensions persist even when distilling larger models. Further discussion of the persistence of outlier dimensions is available in Section C.

## 3.2 Testing Outlier Dimensions for Task-Specific Knowledge

**Methods** In order to test the hypothesis that outlier dimensions contain task-specific knowledge, we attempt to complete an inference task using only the outlier dimension in the fine-tuned model with the highest variance. For the remainder of this paper, we refer to the outlier dimension with the highest variance as the *principal outlier dimension*, which we denote as $\rho$. After fine-tuning our model, we use a simple brute-force algorithm to find a linear decision rule to complete the downstream GLUE task using only the principal outlier

| Task | BERT | ALBERT | DistilBERT | RoBERTa | GPT-2 | Pythia-70M | Pythia-160M | Pythia-410M |
|------|------|--------|------------|---------|-------|------------|-------------|-------------|
| SST-2 | 91.86/77.58 Δ15.54 | 91.90/84.32 Δ8.25 | 90.14/54.39 Δ39.66 | 94.35/63.85 Δ 32.33 | **91.77/91.69** Δ0.09 | 87.30/72.62 Δ16.82 | **89.25/86.64** Δ 2.95 | **94.53/92.19** Δ 2.48 |
| QNLI | 90.11/69.69 Δ22.66 | **91.28/88.16** Δ3.42 | 86.48/66.12 Δ23.54 | 92.78/59.60 Δ 35.67 | **87.80/85.90** Δ2.16 | 80.14/62.93 Δ21.47 | **85.41/82.87** Δ2.97 | **91.41/91.41** Δ0.0 |
| RTE | 61.01/55.60 Δ8.87 | 66.70/63.09 Δ5.41 | **55.14/52.44** Δ4.90 | 76.56/70.31 Δ 8.16 | **61.64/59.74** Δ3.08 | **55.14/53.97** Δ 2.12 | 61.82/50.00 Δ 19.12 | 71.88/ 47.66Δ33.69 |
| MRPC | 84.80/76.04 Δ10.33 | 87.01/79.72 Δ8.38 | 81.56/75.06 Δ7.97 | 86.15/80.70 Δ 6.33 | 78.92/74.08 Δ6.13 | **70.71/68.75** Δ 4.12 | 74.69/68.26 Δ 8.38 | 79.68/ 63.28Δ 20.58 |
| QQP | 90.13/68.27 Δ24.25 | 90.01/85.37 Δ5.15 | 89.12/71.70 Δ19.55 | 90.99/81.53 Δ 10.40 | **89.38/86.91** Δ2.76 | 86.88/70.93 Δ 18.36 | **89.12/85.77** Δ 3.76 | 85.94/80.47 Δ6.36 |
| Avg. | 83.58/69.43 Δ16.33 | 85.38/80.13 Δ6.12 | 80.48/63.94 Δ19.12 | 88.17/71.21 Δ 19.23 | **81.90/79.66** Δ2.85 | 76.23/65.84 Δ 12.58 | 80.06/74.71 Δ 7.48 | 84.69/75.02 Δ11.44 |

Table 1: Comparing the performance of the fully fine-tuned model to our brute-force algorithm on the *principal outlier dimension*, $\rho$ (Equation 1). We compute the percent decrease between the full model performance and the 1-D performance using Equation 1 as full-performance minus 1D-performance divided by full-performance. The reported performance is an average over 4 random seeds.

dimension. We first collect a small sample of 500 sentence embeddings from the training data to find $\rho$ and calculate its mean value, which we denote as $\mu_\rho$. Equation 1 describes the classification decision rule for an input sentence using only $\rho$:

$$x_{\text{label}} = \begin{cases} 0 & \text{if } x_\rho \leq \mu_\rho + \epsilon \\ 1 & \text{if } x_\rho > \mu_\rho + \epsilon \end{cases} \quad (1)$$

where $x_\rho$ denotes the principal outlier dimension for an input sentence $x$, $\epsilon \in \{-50, -49.5, ..., 49.5, 50\}$. Let $\bar{X}_{\text{label}}$ denote the training accuracy of Equation 1. If $1 - \bar{X}_{\text{label}} > \bar{X}_{\text{label}}$ we flip the inequalities in Equation 1:

$$x_{\text{label}} = \begin{cases} 0 & \text{if } x_\rho \geq \mu_\rho + \epsilon \\ 1 & \text{if } x_\rho < \mu_\rho + \epsilon \end{cases}$$

After finding both the value of $\epsilon$ that maximizes accuracy on the training data and the correct direction of the inequality, we measure the ability of $\rho$ to complete downstream tasks using Equation 1 on the hidden validation data.

**Results**    In GPT-2, using the principal outlier dimension results in only a 3% performance drop compared to using the full model representations for most tasks (Table 1). Further, there are several tasks in ALBERT, Pythia-160M, and Pythia-410M where there is little to no change in performance when using only the principal outlier dimension. Although outlier dimensions encode task-specific knowledge in some models, outlier dimensions in BERT, DistilBERT, RoBERTa, and Pythia-70M are insufficient for completing most downstream tasks. In particular, for QNLI, using our brute-force algorithm on a single outlier dimension in GPT-2, ALBERT, Pythia-160M, and Pythia-410M only results in a 2.16%, 3.42%, 2.97%, and 0% performance decrease, where performance on QNLI drops by 22.66%, 23.54%, 33.67% and 21.47% for BERT, DistilBERT, RoBERTa, and Pythia-70M

respectively. Additionally, the average percent decrease in performance is significantly lower for GPT-2 (2.85%), ALBERT (6.12%), Pythia-160M (7.48%) compared to BERT (16.33%), DistilBERT (19.12%) and RoBERTa (19.23%).

### 3.3   Variance vs. 1D-Performance

**Methods**    In this section, we extend our experiment in Section 3.2 by testing each 1-D subspace in model activations for task-specific knowledge. Namely, we use our brute-force algorithm in Equation 1 to learn a linear threshold for each dimension of the model's sentence embeddings.

**Results**    Figure 3 shows multiple 1-D subspaces in model sentence embeddings that contain task-specific information. Importantly, the downstream performance of a single dimension is strongly correlated with the variance in that given dimension. Even in cases where the principal outlier dimension does not contain enough task-specific information to complete the downstream task, there are several non-principal outlier dimensions that are capable of completing the downstream task with a high degree of accuracy. For instance, applying Equation 1 to the largest outlier dimension in BERT on QNLI results in a 22.26% decrease in performance, whereas using Equation 1 on the 5th largest outlier dimension only leads to a 0.56% reduction in performance. We even find a few cases where applying Equation 1 results in *improved* performance over feeding the full representations into a classification head. For full results and further discussion, see Table 3 and Section D in the Appendix.

## 4   Discussion

Previous studies investigating the role of outlier dimensions tend to only focus on BERT or RoBERTa, yet make broad claims about the role of outlier dimensions for LLMs as a whole (Kovaleva et al., 2021; Puccetti et al., 2022). Our results demonstrate that outlier dimensions have different func-

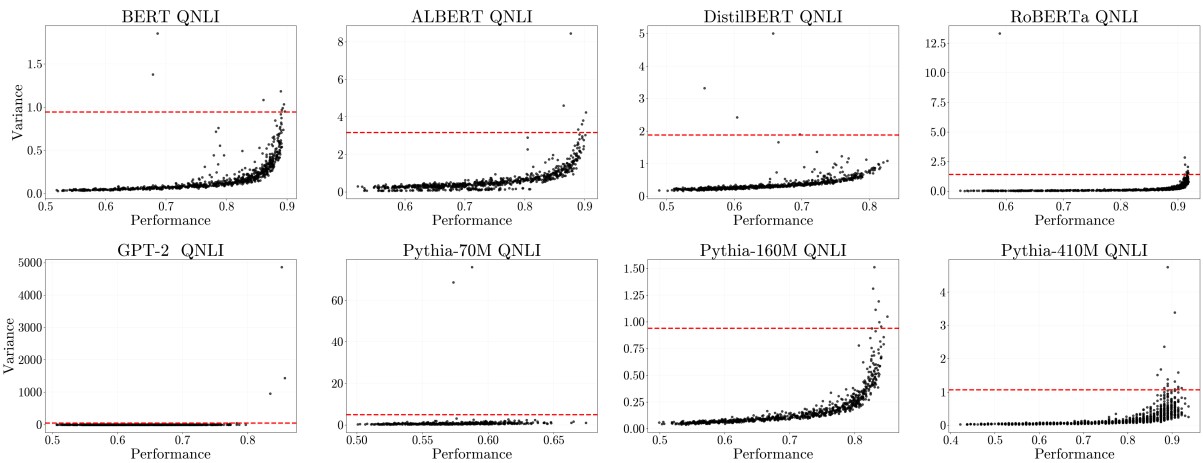

Figure 3: Comparing the downstream performance of all 1-D subspaces of sentence embedding activations on QNLI against the variance in that given dimension. Results for all tasks are available in Section D in the Appendix. The red dashed line indicates the threshold for whether a dimension qualifies as an "outlier dimension" (i.e., 5x the average variance in vector space).

tions for different models and tasks. In particular, outlier dimensions contain task-specific knowledge that can complete downstream fine-tuning tasks for some models (GPT-2, ALBERT, Pythia-160M, and Pythia-410M) but not for others (BERT, Distil-BERT, RoBERTa, and Pythia-70M). Future work should avoid generalizing results from one observation to all other models.

Although numerous studies have argued that outlier dimensions are harmful to model representations (Gao et al., 2019; Timkey and van Schijndel, 2021; Cai et al., 2021), we quantitatively show that the single principal outlier dimension can store enough task-specific knowledge in GPT-2, AL-BERT, Pythia-160M, and Pythia-410M to complete downstream tasks. In cases where the principal outlier dimension does not contain sufficient task-specific knowledge to complete downstream tasks, we find that there are often non-principal outlier dimensions that retain high 1-D performance. In particular, Figure 3 shows that there are non-principal outlier dimensions in BERT and RoBERTa that can complete QNLI using Equation 1 with only a 0.56% and 1.01% performance decrease compared to using full model representations and a classification head. These findings help explain recent results showing that *encouraging* even larger variance in outlier dimensions is beneficial to LLM fine-tuning performance (Rudman and Eickhoff, 2023).

Additionally, our finding that 1-D subspaces contain task-specific knowledge strengthens the arguments that LLMs store linguistic knowledge in a low-dimensional subspace (Hernandez and Andreas, 2021; Coenen et al., 2019; Zhang et al., 2023). The persistence of the same small set of outlier dimensions in pre-training and fine-tuning across various classification tasks and random seeds provides strong evidence that the low-dimensional subspaces learned for different tasks are highly similar. Namely, when fine-tuning, certain LLMs adapt the same small set of outlier dimensions to store task-specific knowledge.

## 5 Conclusions & Future Works

This study challenges the dominant belief in the literature that outlier dimensions are detrimental to model performance by demonstrating that 1) the exact outlier dimensions that emerge in pre-training persist when fine-tuning models regardless of the classification task or random seed and 2) in some LLMs, outlier dimensions contain enough task-specific knowledge to linearly separate points by class label. However, it is still unclear why the principal outlier dimension contains task-specific knowledge in some models and not others. Future work should investigate the specifics of these occurrences and how this finding is affected by model scale, architectural choices, and training objectives. Ultimately, understanding the mechanisms and implications of outlier dimensions in LLMs can contribute to advancements in transfer learning, model interpretability, and optimizing performance in several NLP tasks.

# 6 Limitations

Our study follows the common practice of only considering binary classification tasks. Studies have yet to investigate how changing outlier dimensions behave when fine-tuning for alternative tasks such as question-answering or generative tasks. We limit our analysis to smaller models that are easy to fine-tune. We do not consider how model size impacts the presence of outlier dimensions and whether outlier dimensions store task-specific information with very large LLMs. However, outlier dimensions will likely continue to play a role in larger models, given that outlier dimensions are persistent in GPT-2 and that most of the largest models in NLP are transformer decoders.

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

## A  Dataset Details

Stanford Sentiment Treebank with 2 classes (SST-2) is a binary classification task where models must determine whether a short movie review is positive or negative in sentiment (Socher et al., 2013). Question-answering Natural Language Inference (QNLI) is a binary natural language inference task where models must decide whether or not a given answer is entailed from a specified question (Rajpurkar et al., 2016). Recognizing Textual Entailment (RTE) is a binary classification task where a model must determine if a given sentence logically follows a preceding sentence. The Microsoft Research Paraphrase Corpus (MRPC) tasks models with determining if a pair of sentences are paraphrases of each other (i.e., semantically equivalent). The Quora Question Pairs (QQP) dataset consists of question pairs from Quora. Models must determine if the sentence pairs are semantically equivalent. Note that all tasks are datasets in the GLUE benchmark (Wang et al., 2018).

| Model | L.R. | Batch Size | Epoch |
|---|---|---|---|
| **BERT** | 3e-5 | 32 | 2 |
| **ALBERT** | 1e-5 | 32 | 3 |
| **DistilBERT** | 1e-5 | 64 | 5 |
| **RoBERTa** | 1e-5 | 64 | 3 |
| **GPT-2** | 1e-5 | 32 | 3 |
| **Pythia-70M** | 1e-5 | 32 | 4 |
| **Pythia-160M** | 1e-5 | 32 | 4 |
| **Pythia-410M** | 1e-5 | 32 | 4 |

Table 2: Detailed model hyperparameters.

## B  Hyperparameter Details

Following Geiping and Goldstein (2022), we hyperparameter-tune each model on a single task to learn a set of global hyperparameters. We then use the set of global hyperparameters to fine-tune each

| | BERT | | ALBERT | | DistilBERT | | RoBERTa | | GPT-2 | | Pythia-70M | | Pythia-160M | | Pythia-410M | |
|---|---|---|---|---|---|---|---|---|---|---|---|---|---|---|---|---|
| Task | Acc | Var % | Acc | Var % | Acc | Var % | Acc | Var % | Acc | Var % | Acc | Var % | Acc | Var % | Acc | Var % |
| SST-2 | 91.37 Δ 0.53 | 96 | 91.69Δ0.23 | 99 | 89.42Δ0.80 | 99 | 93.26Δ1.15 | 96 | 91.43Δ0.37 | 99 | 76.03Δ12.91 | 64 | 87.23Δ2.22 | 96 | 94.53 Δ 0.00 | 88 |
| QNLI | 89.66 Δ 0.56 | 99 | 90.23Δ1.15 | 99 | 82.65Δ4.43 | 98 | 91.84 Δ 1.01 | 94 | 85.73Δ2.36 | 99 | 67.46Δ15.82 | 56 | 85.00Δ0.47 | 99 | **93.75 Δ+2.56** | 92 |
| RTE | **61.73Δ+1.18** | 88 | **67.68Δ+1.48** | 98 | **55.69Δ+0.99** | 31 | **79.69Δ+4.09** | 95 | 60.29Δ2.19 | 100 | **55.59Δ+0.83** | 38 | 55.32Δ10.50 | 19 | 60.94Δ15.23 | 70 |
| MRPC | 80.69Δ4.83 | 98 | 82.97Δ4.65 | 92 | 74.08Δ9.17 | 95 | 84.38Δ2.1 | 93 | 75.06Δ4.89 | 99 | 68.38Δ4.64 | 13 | 68.44Δ8.36 | 96 | 70.31Δ11.76 | 87 |
| QQP | 87.59Δ2.81 | 93 | 85.56Δ5.94 | 94 | 81.29Δ8.78 | 97 | 89.14Δ2.03 | 90 | 87.07Δ2.59 | 99 | 70.54Δ18.80 | 66 | 85.17 Δ 4.43 | 70 | **89.84Δ+4.54** | 98 |
| Avg. | 82.21 Δ 1.50 | 97 | 83.67Δ1.90 | 97 | 76.62Δ4.44 | 85 | 87.66Δ0.43 | 94 | 79.91Δ2.48 | 99 | 67.60Δ10.27 | 58 | 76.24Δ5.20 | 76 | 81.88Δ3.9 | 87 |

Table 3: Maximum value of applying our brute force algorithm to all 1D subspaces in the last layer sentence embedding to the full model performance. Δ represents the percent change between the maximum 1D value and the full model performance. Note that bold entries are those where the best 1D subspace performs **better** than the full model representations. Our brute force algorithm on the best 1D subspace. Variance % is the percentile of the variance of the dimension that corresponds to the maximum brute-force classification accuracy. The reported performance is an average over 4 random seeds.

model on the remaining tasks. For this study, we learn global hyperparameters from QNLI and train our models using 4 random seeds. Note that we exclude COLA (Warstadt et al., 2019) from analysis as Geiping and Goldstein (2022) find COLA is highly sensitive to hyperparameter tuning and performs poorly with a global set of parameters. For each model, we search for the optimal combination of learning rate {1e-5, 3e-5, 5e-5, 1e-4}, batch size {16, 32, 64} and the number of training epochs {1,2,3,4,5}. To correctly perform the hyperparameter tuning, we randomly remove 5% of the training data and use it as hyperparameter-tuning evaluation data since we report performance results on the GLUE validation data. Table 2 details the complete set of hyperparameters used for each task in this paper. **Note:** we train all of our models using mixed-precision training, *except* for Pythia-70M and Pythia-160M. For Pythia-70M and Pythia-160M, we use full precision since we received NaN values in our loss function when using mixed-point precision training. Once we learn the global hyperparameters set in Table 2, we fine-tune the models on random seeds 1,2,3,4.

## C Persistence of Outlier Dimensions Continued

First, note that the dimension of Pythia-70M's activation space is 512, and the dimension of Pythia-410M's activation space is 1028. All other models have activation spaces with 768 dimensions. Table 4 lists the number of outlier dimensions and the average maximum variance value of all models on all tasks. Fine-tuning models increases both the number of outlier dimensions present in embedding space as well as the average maximum variance value. This trend is the strongest for the encoder models considered in this paper as well as GPT-2 and Pythia-70M.

| | Pre-Trained | | Fine-Tuned | |
|---|---|---|---|---|
| Model | Num Outliers | Avg. Var($\rho$) | Num Outliers | Avg. Var($\rho$) |
| **BERT** | 2 | 0.10 | 62 | 4.87 |
| **ALBERT** | 3 | 3.74 | 60 | 9.30 |
| **DistilBERT** | 3 | 0.04 | 25 | 4.68 |
| **RoBERTa** | 4 | 0.02 | 64 | 13.45 |
| **GPT-2** | 6 | 619.18 | 8 | 3511.82 |
| **Pythia-70M**[*] | 2 | 13.49 | 4 | 61.54 |
| **Pythia-160M** | 35 | 32.71 | 27 | 9.85 |
| **Pythia-410M**[*] | 11 | 12.15 | 129 | 6.48 |

Table 4: Counts of the number of outlier dimensions that appear across all tasks and the average variance value of the principal outline dimensions, $\rho$.

## D All 1-D Results

**Variance vs. Performance** In this Section, we provide full results for the experiment in Section 3.3. Namely, we apply Equation 1 to all 1-dimensional subspaces in the sentence embeddings of each model on every task. For nearly every model and every task, we find the same strong correlation between a dimension variance and the ability to encode task-specific information. There are very few exceptions to this trend. However, there are two tasks (RTE& MRPC) where there is no strong correlation between performance and variance for some of the models considered in this paper. Note that in MRPC, a correlation between high variance value and performance does not emerge as Pythia-70M and Pythia-160M barely perform above predicting the class majority label of 68.38%. In fact, no 1-D subspace of Pythia-70M fine-tuned on MRPC performs above "random" performance of 68.38%. For RTE, there is only a mild correlation between variance and performance for BERT, ALBERT, and RoBERTa. For DistilBERT and the Pythia models, the correlation between variance and performance degrades even further. We hypothesize this is in part due to RTE being a difficult task where models tend to perform poorly. An interesting direction of future work would be to see if there are types of tasks, such as question-answering

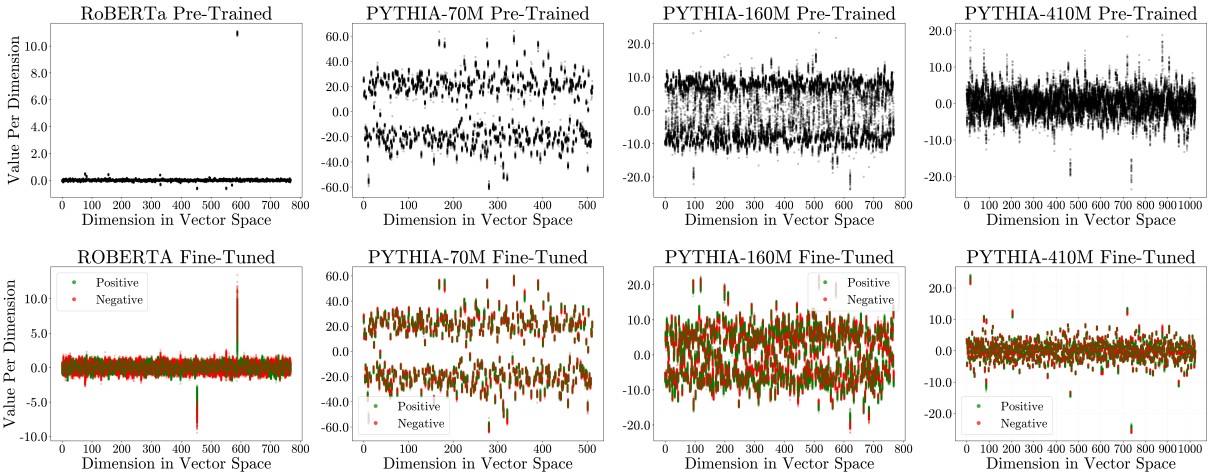

Figure 4: Average activation diagrams of sentence embeddings on the SST-2 validation dataset. The x-axis represents the index of the dimension, and the y-axis is the corresponding magnitude in that given dimension. Top: pre-trained models where no fine-tuning occurs. Bottom: models fine-tuned to complete SST-2.

or textual entailment, that impact an outlier dimension's ability to store task-specific information.

**Maximal 1-D Subspaces** Figures 3 and 5 demonstrate that oftentimes the principal outlier dimension is not the best performing 1-D subspace. In Table 3, we report the *maximum* performance of a 1-D subspace after applying Equation 1 to complete the downstream task along with the percentile of the variance of the maximal 1-D subspace. Trends in Table 3 provide further evidence for our finding that the variance of the activation value correlates with 1-D performance. With the exception of Pythia-70M, the average performance decrease between the maximum 1-D performance and the full model performance is less than ≈5% for all models considered in this paper. Surprisingly, we find 7 cases where the maximum 1-D subspace performs *better* than feeding the full representations into a classification head. This finding is the most pronounced on RoBERTa-RTE, Pythia-410M-QQP, and Pythia-410M-QNLI, where the best 1-D subspace improves upon full model performance by 4.09%, 4.54%, and 2.56%, respectively.

## E All Activation Diagrams

In this section, we report the remaining models' (RoBERTa, Pythia-70M, Pythia-160M, and Pythia-410M) activation diagrams on SST-2. Trends on SST-2 are representative of all of the tasks considered in this paper. We report a similar phenomenon of variance drastically increasing in RoBERTa and Pythia-70M after fine-tuning, particularly in outlier dimensions. The Pythia models, however, exhibit different trends. In Pythia-160M and Pythia-410M, the average variance in the principal outlier dimension decreases after fine-tuning. Table 4 shows that the average max variance decreases from 32.71 to 9.85 in Pythia-160M and decreases from 12.15 to 6.48 in Pythia-410M. Interestingly, in Pythia-70M and Pythia-160M, the embedding dimensions are much further from the origin compared to every other model considered in this paper. Our results highlight how, even for models with similar architectures (Pythia models and GPT-2), the structure of embedding space can be very dissimilar. Further research is needed to understand how model architecture and training objectives impact the structure of model embeddings.

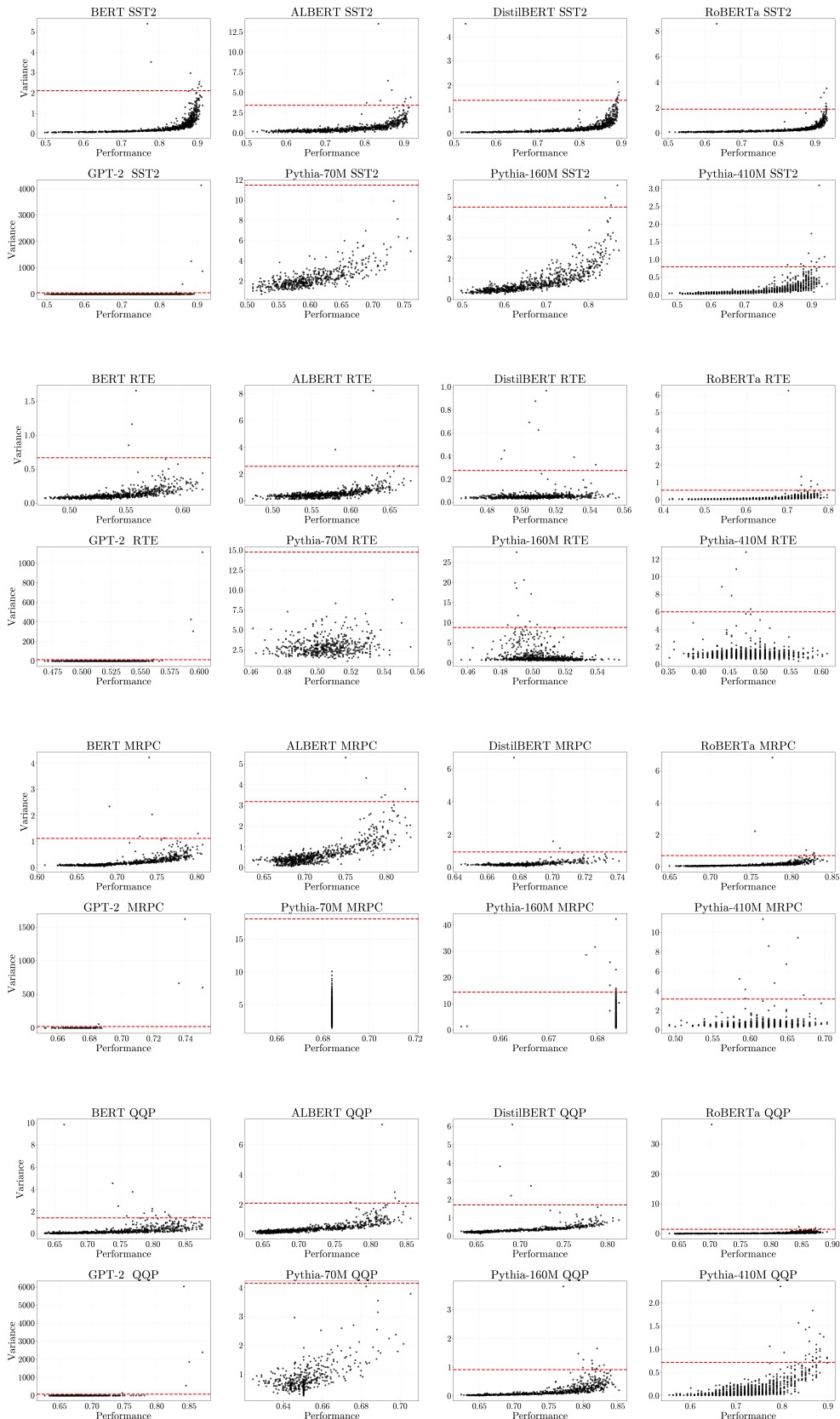

Figure 5: Comparing the downstream performance of all 1-D subspace of sentence embedding activations on SST-2, RTE, MRPC, and QQP against the variance in that given dimension. Results for all tasks are available in Section D in the Appendix. The red dashed line indicates the threshold for whether a dimension qualifies as an "outlier dimension" (i.e. 5x the average variance in vector space).