# OpenReview forum: "Outlier Dimensions Encode Task Specific Knowledge"
_EMNLP/2023/Conference — EMNLP 2023 Main_

### Official Review · Reviewer_BqrZ · 2023-07-22

**Soundness:** 4

**Excitement:**

4: Strong: This paper deepens the understanding of some phenomenon or lowers the barriers to an existing research direction.

**Paper Topic And Main Contributions:**

Content: The paper presents a study on how outlier dimensions of word representation are preserved in fine-tuning, and how efficient they are in down-stream tasks using LLMs compared to BERTs.

**Questions For The Authors:**

Question 1: L 165: is 3511.82 correct here? It looks as it would be 300 or something?

Question 2: Can we draw any conclusion about language characteristics from the fact that only a couple of outlier dimension are sufficient for downstream tasks using LLMs? That would be really interesting.

Question 3: Spelling out the task abbreviations would be helpful.

Question 4: A couple of more words on what outlier dimensions are would suffice.

**Reasons To Accept:**

Positive: The paper shows that a linear threshold on only a few outlier dimensions decreases performance in downstream tasks for large language models only a little bit. it is written clearly and points out important characteristics for outlier dimensions.

**Reasons To Reject:**

see Questions

**Reproducibility:**

4: Could mostly reproduce the results, but there may be some variation because of sample variance or minor variations in their interpretation of the protocol or method.

**Reviewer Confidence:**

4: Quite sure. I tried to check the important points carefully. It's unlikely, though conceivable, that I missed something that should affect my ratings.

---

> ### Author Rebuttal · Authors · 2023-08-28
>
> Thank you for your thorough review and insightful questions.
>
> $\textbf{Question 1:}$  We apologize for the lack of clarity. Figure 1 reports the average sentence embeddings across different random seeds. Note that the average variance across random seeds will not, in general, equal the variance of the average of sentence embeddings. In contrast to Figure 1, the values we report in L 165 are the average of the variance, not the variance of the average. Further, the results reported in L165 are an average of all tasks and all random seeds compared to Figure 1. which visualizes the average sentence embeddings across four random seeds on SST-2.
>
> $\textbf{Question 2:}$  That is a great question! It is also a hard one to answer satisfactorily. Previous works (Gao et al. 2019, Pucceti et al. 2022) have connected outlier dimensions to a  token imbalance in the pre-training corpus, with Gao et al. 2019 suggesting that the Zipf distribution of word frequencies causes outlier dimensions. Upon receiving reviews, we again examined the token distribution frequency in the fine-tuning datasets. For SST-2, we find that certain words that are highly correlated with a given label of positive or negative sentiment, such as,  “good”,  “funny”, “bad,” and “dull”, occur with very high frequency in the SST-2 training data. Although this is beyond the scope of our study, a fruitful direction for future work would be to examine words in the training data with the highest mutual information and frequency with a given class label and examine the impact on 1D performance and the embedding norm of sentence embedding by swapping these words with less frequent synonyms that should not alter the model’s prediction. Concretely, we could replace the word “funny” in a sentence in SST-2 with a less common synonym, such as “chucklesome,” and measure the impact on outlier dimension variance and 1D performance.
>
> $\textbf{Question 3:}$ Apologies for this oversight. We will provide the full names of the datasets in the paper and briefly describe each task in the appendix.
>
> $\textbf{Question 4:}$ To address this, we will amend the description of “outlier dimensions” in our paper: Outlier dimensions are dimensions in model representations where the variance and magnitude of the dimension are significantly larger than the rest of the embedding space. Following Kovaleva et al. 2021 and Pucetti et al. 2022, a given dimension is an outlier dimension if the variance of the dimension is at least 5x larger than the average variance in the embedding space. Namely, given a set of sentence embeddings $X \in R^{n}$, with covariance matrix $\Sigma$ a dimension $i \in \{1,..,n\}$ is an outlier dimension if $\Sigma_{i,i} \geq 5 \cdot \frac{1}{n} \sum_{k=1}^{n} \Sigma_{k,k}$

---

### Official Review · Reviewer_Qjdk · 2023-08-05

**Soundness:** 4

**Excitement:**

4: Strong: This paper deepens the understanding of some phenomenon or lowers the barriers to an existing research direction.

**Paper Topic And Main Contributions:**

This paper studied the role of outlier dimensions (dimensions of much larger magnitude) in sentence embeddings (obtained from pre-trained language models). Despite previous studies argue that outlier dimensions hurt the quality of sentence embeddings, the authors observed that these outlier dimensions encoded important task-specific information, because a single outlier dimension sometimes (i.e., on some models/tasks) can perform downstream tasks with little performance loss. Moreover, the authors obtained a few interesting observations, including 1) outlier dimensions from pre-training phrase presisted after fine-tuning, and 2) outlier dimensions are shared before and after distillation (by comparing BERT and DistillBERT).

**Questions For The Authors:**

What do you think these results can derive, in terms of future sentence embedding pre-training design? The paper will also be strengthen by adding this discussion.

**Reasons To Accept:**

* Very interesting results which are different from previous common wisdom that outlier dimensions are harmful.
* Insightful observation that outlier dimensions encode task-specific tasks by a simple yet intriguing linear classification experiment.
* The paper is easy-to-follow and well-written.

**Reasons To Reject:**

* Section 3.1 studied the maginitude of outlier dimensions; however, only shallow observations were offered, without further analyses or insights.
* In experiments, the authors compared different models, and observed a few interesting differences between different architectures. However, the paper will be stronger if it can connect these models’ training objectives/architecture choices with these differences. Despite this, I understand that this can be very difficult and hard to elaborate on in a short paper. This is more like a wish.

**Reproducibility:**

4: Could mostly reproduce the results, but there may be some variation because of sample variance or minor variations in their interpretation of the protocol or method.

**Reviewer Confidence:**

4: Quite sure. I tried to check the important points carefully. It's unlikely, though conceivable, that I missed something that should affect my ratings.

---

> ### Author Rebuttal · Authors · 2023-08-28
>
> Thank you for your questions and your thorough review of our paper!
>
> Our results provide some evidence that an individual dimension's ability to complete a downstream task correlates with the magnitude of the individual dimension. Namely, the fact that the average variance value of the principal outlier dimensions in GPT-2 (3511.82) and ALBERT (9.30) are significantly larger than the principal outlier dimensions in BERT (4.87) and DistilBERT (4.68) is likely related to the ability of that dimension to store task-specific information. After running additional experiments, we find a strong correlation between variance and 1D performance. We notice that for BERT and DistilBERT, it is not always the single largest outlier dimension that encodes most task information. However, high variance dimensions are much more likely to cover critical task knowledge. Applying Equation 1, to learn a linear threshold on the 6th largest outlier dimension in BERT fine-tuned on SST-2 yields a classification accuracy of 90.73% compared to a classification accuracy of 91.86%  on the full model representations. The revised version of the paper will include detailed graphs and a dedicated discussion of this relationship.
>
>  We cannot draw definitive conclusions as to whether differences in training objectives and model architecture play a role in determining whether the principal outlier dimension contains class-specific information. However, it is interesting to note that the variance of GPT2’s principal outlier dimension is much greater than those of its BERT-based counterparts. We are currently running experiments with an additional transformer encoder architecture: RoBERTa (Liu et al. 2019) and several transformer decoder architectures contained in the Pythia  (Biderman et al. 2023) suite of models to better understand how different model architectures and pretraining objectives impact a model’s ability to store class-specific information into outlier dimensions and to test further the hypothesis that the variance in a given outlier dimension correlates to improved downstream performance. Preliminary findings show that outlier dimensions in Pythia-70M, Pythia-160M, and Pythia-410M all encode task-specific information similarly to GPT-2.

---

### Official Review · Reviewer_WcPR · 2023-08-10

**Soundness:** 3

**Excitement:**

3: Ambivalent: It has merits (e.g., it reports state-of-the-art results, the idea is nice), but there are key weaknesses (e.g., it describes incremental work), and it can significantly benefit from another round of revision. However, I won't object to accepting it if my co-reviewers champion it.

**Paper Topic And Main Contributions:**

This paper investigates the presence and impact of outlier dimensions in representations from language models  during fine-tuning for downstream tasks.

The main contributions of the paper are as follows:

The paper explores the phenomenon of outlier dimensions in representations from LLMs. It highlights that these outlier dimensions dominate the representation space and are characterized by high variance.

The paper demonstrates that outlier dimensions are crucial for downstream tasks, as ablating or removing them hurts the downstream task performance. This suggests that outlier dimensions encode task-specific knowledge that is important for the success of the fine-tuned models.







**Reasons To Accept:**

The paper explores a novel aspect of language models by focusing on outlier dimensions and their impact on both pre-trained and fine-tuned models. This unique perspective adds to the understanding of how language models operate and influence downstream tasks.

The study finds that in certain LLMs (GPT-2 and ALBERT), a single outlier dimension alone can perform well in completing downstream tasks, achieving a minimal error rate. This implies that a single dimension can drive the decision-making of the downstream models.

**Reasons To Reject:**

The study shows that the beneficial influence of outlier dimensions might be specific to the tested models (GPT-2 and ALBERT). It's essential to address why these findings cannot be generalized to other models and architectures.

**Reproducibility:**

4: Could mostly reproduce the results, but there may be some variation because of sample variance or minor variations in their interpretation of the protocol or method.

**Reviewer Confidence:**

4: Quite sure. I tried to check the important points carefully. It's unlikely, though conceivable, that I missed something that should affect my ratings.

---

> ### Author Rebuttal · Authors · 2023-08-28
>
> Thank you for your insightful comments. Our study shows that outlier dimensions are beneficial specifically for some models (GPT-2 and ALBERT) and not others (BERT and DistilBERT), which is different from previous claims that outlier dimensions are harmful to LLMs in general (Kovaleva et al. 2021, Puccetti et al. 2022). Given the similarities between ALBERT and BERT/DistilBERT, such as the same vocabulary size, pre-training task, GELU activation functions, and transformer encoder architecture, it is difficult to determine the exact factors that cause a reliance on outlier dimensions. There are, however, a few notable architectural differences between ALBERT and BERT/DistilBERT, such as the use of parameter sharing in ALBERT. To determine which architectural features allow a model to store class information in outlier dimensions, we are actively expanding our results to another prominent decoder model in the literature: RoBERTa (Liu et al. 2019). GPT-2 is the only decoder model and, thus, was the only model trained with a language modeling objective. We are currently running experiments using a selection Pythia (Biderman et al. 2023) transformer decoder models to test if our findings hold for a wider variety of transformer decoder models. Preliminary findings show that outlier dimensions in Pythia-70M, Pythia-160M, and Pythia-410M all encode task-specific information similarly to GPT-2, indicating that model architecture and pre-training objectives are highly important in a model’s ability to store task-specific information into a single dimension.
>
> Our additional experiments will allow us to make more generalizable conclusions about the role of outlier dimensions in fine-tuning and will help provide more evidence as to what specific architectural or pre-training task differences allow our results to generalize to other models.

---

### Meta-Review · Area_Chair_Zmca · 2023-09-17

**Recommendation:** 5

**Metareview:**

The paper investigates the impact of fine-tuning on outlier dimensions providing two insights. First, outlier dimensions in pretraining persist in fine-tuned models and second, a single outlier dimension can provide reasonable performance on a downstream task. The paper provides a nice novel insight that is interesting and well-presented.

Proposed points of improvement are (1) that the aspect that results may not generalise should be discussed: the rebuttal provides additional experiments and mentions similarities between models. (2) Analyses remain shallow and (3) address what conclusions can be drawn from the observation that a couple of outlier dimensions are sufficient.

It can not be expected of a short paper, where an interesting observation that leads to new research directions is sufficient, to address all of the above. Given sufficient space, it would indeed be important to discuss what can be expected in terms of generalization and what remains for future research (if it only applies to some LMs, it's interesting as well) and what it means that even one outlier dimension can be fine-tuned for a task: should that even be the case?

---

### Decision · Program_Chairs · 2023-10-07

**Decision:**

Accept-Main

**Comment:**

The paper investigates the impact of fine-tuning on outlier dimensions providing two insights. First, outlier dimensions in pretraining persist in fine-tuned models and second, a single outlier dimension can provide reasonable performance on a downstream task. The paper provides a nice novel insight that is interesting and well-presented.

Proposed points of improvement are (1) that the aspect that results may not generalise should be discussed: the rebuttal provides additional experiments and mentions similarities between models. (2) Analyses remain shallow and (3) address what conclusions can be drawn from the observation that a couple of outlier dimensions are sufficient.

It can not be expected of a short paper, where an interesting observation that leads to new research directions is sufficient, to address all of the above. Given sufficient space, it would indeed be important to discuss what can be expected in terms of generalization and what remains for future research (if it only applies to some LMs, it's interesting as well) and what it means that even one outlier dimension can be fine-tuned for a task: should that even be the case?